# Application of High-Speed Quantum Cascade Detectors for Mid-Infrared, Broadband, High-Resolution Spectroscopy

**DOI:** 10.3390/s21175706

**Published:** 2021-08-24

**Authors:** Tatsuo Dougakiuchi, Naota Akikusa

**Affiliations:** 1Central Research Laboratory, Hamamatsu Photonics K.K., 5000 Hirakuchi, Hamakita-ku, Hamamatsu City 434-8601, Japan; 2Laser Promotion Division, Hamamatsu Photonics K.K., 5000 Hirakuchi, Hamakita-ku, Hamamatsu City 434-8601, Japan; aki@lpd.hpk.co.jp

**Keywords:** mid-infrared, quantum cascade detector, high-speed operation, heterodyne detection, high-resolution spectroscopy

## Abstract

Broadband, high-resolution, heterodyne, mid-infrared absorption spectroscopy was performed with a high-speed quantum cascade (QC) detector. By strictly reducing the device capacitance and inductance via air-bridge wiring and a small mesa structure, a 3-dB frequency response over 20 GHz was obtained for the QC detector, which had a 4.6-μm peak wavelength response. In addition to the high-speed, it exhibited low noise characteristics limited only by Johnson–Nyquist noise, bias-free operation without cooling, and photoresponse linearity over a wide dynamic range. In the detector characterization, the noise-equivalent power was 7.7 × 10^−11^ W/Hz^1/2^ at 4.6 μm, and it had good photoresponse linearity up to 250 mW, with respect to the input light power. Broadband and high-accuracy molecular spectroscopy based on heterodyne detection was demonstrated by means of two distributed-feedback 4.5-μm QC lasers. Specifically, several nitrous oxide absorption lines were acquired over a wavelength range of 0.8 cm^−1^ with the wide-band QC detector.

## 1. Introduction

Quantum cascade (QC) photovoltaic infrared photodetectors are based on intersubband transitions of electrons [1,2]. Unlike quantum-well infrared photodetectors [3], the active regions of QC detectors can be engineered with highly flexibility, as shown for QC lasers [4]. Consequently, a variety of active region designs have been reported that expand the operational wavelength range or improve the responsivity [5,6,7]. Hence, photoresponses in QC detectors have been demonstrated from the near- to the far-infrared wavelengths [8,9]. QC detectors also exhibit low-noise and high-speed. Due to their bias-free operation, dark currents induced by external voltages are absent, which is important for high detectivity without an elaborate cooling system. Regarding high-speed operation, an electron transit time of less than 1 ps was substantiated in a near-infrared QC detector by a time-resolved pump–probe measurement [10]. More recently, frequency responses of several tens of gigahertz and picosecond response times have been demonstrated in the mid-infrared (MIR) QC detectors [11,12]. The intrinsic short response times are determined by high-speed electron transport via sub-picosecond intersubband scattering processes. Low-noise, high-speed QC detectors could be key devices for high-speed MIR applications.

The MIR is a molecular finger print region and, thus, is very important in fundamental science, medicine, and industry. A large number of unique and strong absorption lines that correspond to fundamental vibrational modes for many molecules are in the MIR, and they can be used to identify and quantify specific molecules. In particular, laser absorption spectroscopy (LAS) in the MIR is a powerful tool for high-precision and high-sensitivity molecular sensing because of the strong absorption of narrow-linewidth laser light [13,14,15,16]. Its most prominent application is the detection and identification of gas-phase molecules, because significantly high sensitivity can be obtained by long-range propagation of a collimated beam in a gas. The atmospheric windows in the MIR enable environmental measurements in free space [17,18]. As multiple absorption lines of several gases can overlap in a specific spectral region, high-resolution, high-sensitivity broadband is always required in LAS.

Heterodyne detection is an established technique to obtain high-resolution in LAS [19], and by using a high-speed detector, broadband measurements can be achieved simultaneously. In the MIR application field, HgCdTe (MCT) detectors are the most widely used photodetectors because of their high-responsivity and broad responsive wavelength covering a few micrometers [20], unlike that of QC detectors. However, in MCT detectors, a high-speed operation of several tens gigahertz is difficult in principle and not adequate for broadband heterodyne detection. On the other hand, the frequency range of a heterodyne beat signal up to 20–30 GHz, the upper limit of practical processers, corresponds to the wavelength tuning range of ~1 cm^−1^. In this point of view, the narrow response spectrum of QC detectors is not concern for heterodyne spectroscopy and is a preferable property to avoid background noises. Here, we demonstrate broadband heterodyne LAS by using QC lasers and a QC detector over a spectral range centered at 4.5 μm. To enhance the high speed of the QC detector, we reduced the parasitic capacitance and inductance with air-bridge wiring and a small mesa structure for the thick active region constructed from 90 cascade modules. A 3-dB cutoff frequency was measured for over 20 GHz, and the wide-band frequency response guaranteed a 0.8 cm^−1^ broadband spectral range for heterodyne spectroscopy. Several absorption lines of nitrous oxide (N_2_O) were observed over the range 2220.59–2219.76 cm^−1^ with 5 MHz resolution.

## 2. Characterization of a Quantum Cascade Detector

A schematic of the conduction band of the QC detector with a coupled-quantum-well design is shown in Figure 1a [7,11,21]. The response wavelength was determined by the energy separation between levels 7 and 1 (*E*_71_ = 289 meV). An incident light that has an energy corresponding to *E*_71_ associated with the electron excitation was absorbed. Due to the asymmetric conduction-band potential, the excited electrons were transferred in a preferential direction in line with the step-like energy levels formed in the sequential quantum wells. In the coupled-quantum-well design, the center of the wavefunction for the upper absorption level 7 was slightly shifted from that of level 1 to the thin well side. Consequently, longer longitudinal optical phonon scattering times (τ_71_~3 ps) were obtained, while the dipole length (*d*_71_~1.1 nm) remained almost unchanged. Additionally, reverse currents caused by electron transitions from levels 6–4 to 1 were prevented by the spatial separation via the thin well between the absorption and transport regions. As the level 7 wavefunction extended to the transport region and overlapped with that of level 6, photoexcited electrons could escape from the absorption region across the thin well and thick barrier.

All of the layer structures consisting of 90 cascade modules were grown on a semi-insulating InP substrate via metal–organic vapor-phase epitaxy. The wafer was processed into a 25-μm-wide mesa stripe and cleaved to a 100-μm length. The cleaved facet was used as the acceptance surface for strong absorption of the incident light propagating along the stripe direction. Both the thick active region of the 90 cascade modules and the narrow 25 μm × 100 μm mesa were essential to reducing the parasitic capacitance. Furthermore, to cut the device inductance, air-bridge wiring was used for electrical connection to the signal-output electrode. The device capacitance was 0.19 pF, as determined with a *C*-meter (4280A, Hewlett-Packard, Palo Alto, CA, USA), and the inductance was 0.21 nH, as calculated from the geometry of the air-bridge wiring [11]. The 3-dB cutoff frequency was estimated to be ~23 GHz, based on an equivalent circuit model, including the 1-ps ultimate electron transition time across one cascade module [11]. The experimental confirmation of the frequency response of the QC detector is shown later.

Figure 1b shows a response spectrum of the uncooled QC detector obtained with a Fourier-transform infrared spectrometer (Nicolet 8700, Thermo Fisher Scientific, Waltham, MA, USA). The peak response wavelength was 4.6 μm (2160 cm^−1^, 267 meV), and *E*_71_ was less than the calculated value of 287 meV. The reason for this difference was because of the insufficient band offset for the higher 7 and 8 energy levels. The energy difference for the second peak, which appeared at a shorter wavelength region, was ~30 meV because of weaker quantum confinement near the top of the barrier height. This difference between design and experiment could be reduced by applying a strain-compensated condition in the InGaAs/InAlAs or by using other wide-bandgap materials, as in shorter-wavelength QC detectors [8,10].

The dark voltage–current characteristics measured at various temperatures over 220–300 K, with 20 K intervals, are shown in Figure 1c. An asymmetric behavior originating from the band structure was observed. Figure 1d is an Arrhenius plot of the differential resistances at zero bias. The estimated activation energy of the device was 251 meV, which corresponded to the transition energy between *E*_61_ (264 meV) and *E*_51_ (228 meV). This indicated that the dark current induced by the transition from the ground level 1 to the 5–2 levels in Figure 1a was suppressed in the coupled-quantum-well design.

Figure 2a plots the output photocurrent as a function of the incident light power. The incident light was the 2220 cm^−1^ continuous wave (CW) distributed-feedback QC laser described below. The photoresponse exhibits good linearity with the incident light power up to 250 mW; the slope derived from the linear fit was estimated to be 4.7 mA/W for the specific wavelength of 2220 cm^−1^ (without compensation for the coupling losses from the focusing lens, surface reflections of the optics, and the cleaved facet acceptance area of the QC detector). Simultaneously, the peak responsivity at 2160 cm^−1^ was 5.7 mA/W, determined from the ratio of the signal intensities between the two wavelengths in the response spectrum in Figure 1b. Figure 2b plots the current noise power spectrum density of the QC detector at room temperature, obtained with a low-noise current amplifier (LCA-40K-100M, FEMTO, Berlin, Germany) and an audio analyzer (SR1, Stanford Research Systems, Sunnyvale, CA, USA). At frequencies higher than 100 Hz, the measured flat noise level matched the calculated Johnson–Nyquist noise level for an 89-kΩ device resistance. Due to the excellent low noise in bias-free operation, the detectivity was improved despite the low responsivity relative to other MIR detectors. The calculated noise-equivalent power was 7.7 × 10^−11^ W/Hz^1/2^, with a peak responsivity and flat noise level of 4.4 × 10^−13^ A/Hz^1/2^.

## 3. Broadband Heterodyne Spectroscopy

The high-speed QC detector was used to acquire broadband heterodyne absorption spectra of N_2_O with the optical setup depicted in Figure 3. A heterodyne beat signal was generated with two identical 4.5 μm distributed-feedback QC lasers (L12004-2209H-C, HAMAMATSU PHOTONICS, Hamamatsu, Japan) [11,21,22]. The emissions from the lasers were collimated with aspheric lenses and combined in a beam splitter for collinear propagation and to focus on the acceptance surface of the QC detector via an optical isolator (MESOS optical isolator, Electro-Optics Technology, Traverse City, MI, USA). For use as a local oscillator, the wavelength of one QC laser (Fixed QC laser in Figure 3) was stabilized at a locked heatsink temperature and a fixed injection current supplied by a low-noise current driver (C16174-01, HAMAMATSU PHOTONICS, Hamamatsu, Japan). The emission wavelength of the “Tuned-QC laser” (Figure 3) was modulated with a ramp wave controlled by a function generator (FGX-2220, TEXIO, Yokohama, Japan). To observe narrow N_2_O absorption lines, a multi-pass cell (2.4-PA, Infrared Analysis, Anaheim, CA, USA) with a 2.4 m optical path length was used in the beam path of the tuned QC laser. The pressure of the N_2_O enclosed in the multi-pass cell was controlled with a vacuum gauge (not shown in Figure 3). The scanning beat signal was detected by the QC detector and accumulated in the spectrum analyzer. The N_2_O absorption lines were observed as extinctions of the signal intensity associated with the wavelength modulation.

### 3.1. Control and Measurement of Beat Signals

Figure 4 shows the CW current–voltage–light output characteristics of the fixed and tuned QC lasers measured at locked heatsink temperatures of 28 °C and 27 °C, respectively. The temperatures were carefully determined for an expedient scanning range of the heterodyne beat signal that included several N_2_O absorption lines, as described in the next section.

The injection current of the fixed QC laser was set to 625 mA to maintain the 2220.6 cm^−1^ emission wavelength, as shown in Figure 5a, where the Fourier-transform infrared spectrum was calibrated with a N_2_O absorption line. To generate a wide-band beat signal, the injection current of the tuned QC laser was varied in a mode-hop-free manner, as exhibited in Figure 5a,b. The frequency of the heterodyne beat signal was tuned from 40 MHz to 26 GHz by varying the injection current of the tuned QC laser over the range 407–494 mA. The frequency of the heterodyne beat signal could be calculated from the difference in wavelengths of the fixed and tuned QC lasers. In Figure 5b, the beat frequency was thus calculated from the fixed and the varied wavelengths for a certain injection current, with good agreement with the measured beat frequencies. Hence, broadband heterodyne spectroscopy could be performed with the well-controlled beat signal over 25 GHz.

The frequency response of the QC detector shown in Figure 6 was measured with the heterodyne setup (Figure 3), without the multi-pass cell, and recorded in the max-hold trace mode of the spectrum analyzer (N9000B, KEYSIGHT TECHNOLOGIES, Santa Rosa, CA, USA). The result was normalized at 0 dB by using the average of the data below 3 GHz, and the signal level was maintained over 35 dB for the entire the frequency range. The theoretical curve based on an equivalent circuit model [11] is also exhibited in Figure 6. Due to the reductions in the parasitic capacitance and inductance, a 3-dB cutoff frequency over 20 GHz was confirmed. The experimental data differed with the theoretical curve at around 25 GHz because of a small impedance mismatch between the device and the measurement system in the high-frequency range. Such a behavior of the frequency response was possible to clear up with measurements over 30 GHz; however, this was limited by our instrument. The optimization of a QC detector for radio-frequency operation was reported in Ref. [12], where a well-designed coplanar waveguide was used to match the 50 Ω impedance, and no noticeable artifacts appeared in the frequency response up to 50 GHz. However, for broadband heterodyne spectroscopy, the variation of the beat signal intensity up to 25 GHz can be regarded as within the 3-dB cutoff.

### 3.2. Observation of N_2_O Absorption Lines

Broadband heterodyne absorption spectroscopy of N_2_O was demonstrated by adding 50 Pa of N_2_O to the multi-pass cell. To reduce the data volume, the spectral acquisition was performed over three separate frequency regions that could be covered with the tuned QC laser while the wavelength of the fixed QC laser was constant. Each measurement region had a 2 GHz width and a 5 MHz resolution bandwidth (they are arbitrarily conditions specified with the spectrum analyzer), and the absorption spectra were recorded for four minutes in the max-hold trace mode of the spectrum analyzer. During data accumulation, the wavelength of the tuned QC laser was repeatedly modulated with the ramp current pulse at a 1 kHz repetition rate and an amplitude of 4 mA for the wavelength sweep.

Figure 7 plots the heterodyne spectroscopy of the N_2_O absorption lines. Figure 7a shows the entire measurement range, while Figure 7b–d were enlarged spectra. Each absorption signal was extracted by using the background obtained with the empty multi-pass cell. The black dots and red lines correspond to the experimental and calculated results, respectively (SpectraPlot [23,24]), where both were normalized by maximum values around 2220.4 cm^−1^. The experimental absorption strengths, peak positions, and spectral widths in the three spectra agree well with the calculations in the spectral range of ~0.8 cm^−1^.

## 4. Discussion

The operational bandwidth of the QC detector was theoretically predicted to be greater than 150 GHz [11] because of the picosecond electron transit times in the QC structures [25,26]. Here, the QC detector response speed was limited by the parasitic capacitance and circuit inductance. Further improvement is required while maintaining the responsivity. The normal incidence schemes have the potential to overcome this challenge by simultaneously enlarging the acceptance area and reducing the capacitance with a smaller device structure [27,28,29,30]. Furthermore, this approach would resolve the polarization dependency of the photoresponse and would make it possible to realize an arrayed QC detector for imaging [31]. However, MIR point-detectors with operational bandwidths of several tens gigahertz are expected to enable innovative technologies. One example would be high-speed spectroscopy [32], based on the strong molecular interaction with MIR light, especially for spectroscopic analysis of instantaneous and non-invertible phenomena, such as explosions, combustion, or chemical reactions. Another example would be free space optical communications that take advantage of the low propagation losses of MIR due to weak absorption in the atmospheric window, and low Rayleigh scattering relative to that in the visible and near-infrared regions [33,34]. QC detectors are highly suitable receivers for any device that requires high-speed communications because they do not consume any power.

Here, broadband heterodyne spectroscopy was demonstrated by using the wide-band QC detector. Although the data acquisition was performed over three separate spectral regions, the entire 1 cm^−1^ tuning range was continuously covered by using a large modulation amplitude for the QC laser injection current and by using optimal resolution. The resolution bandwidth could be optimized for increased resolution and shorter measurement times. Here, they were specified at 5 MHz and 4 min, respectively.

The heterodyne optical system of QC lasers and detectors was a compact spectroscopic module [35,36] for broadband high-resolution gas sensing in cases where absorption lines of several gases are mixed. By combining broadband tunable QC lasers in a CW mode, such as an external cavity configuration [37,38,39,40], for signal and local oscillators, the observable spectral range could be significantly extended [41]. Broadband heterodyne detection would also be essential for absolute frequency determinations when MIR optical frequency combs are used [42]. Wide-band QC detectors and their use in broadband heterodyne optical systems have much potential to expand the application of the MIR spectroscopy.

## 5. Conclusions

High-speed operation with a 3-dB bandwidth over 20 GHz was realized in an uncooled QC detector with a 4.6-μm peak response. This was accomplished by reducing the parasitic capacitance and circuit inductance with a narrow 25 μm × 100 μm ridge-mesa structure, a thick active region of 90 cascade modules, and air-bridge wiring. A noise-equivalent power of 7.7 × 10^−11^ W/Hz^1/2^ was obtained with a peak responsivity of 5.7 mA/W, and a flat noise level of 4.4 × 10^−13^ A/Hz^1/2^. By using the high-speed QC detector, a broadband heterodyne absorption spectrum of N_2_O was obtained over the range of ~0.8 cm^−1^, with a resolution bandwidth of 5 MHz. The absorption strengths, spectral positions, and spectral widths of the absorption lines were in good agreement with the calculations. In future, a compact high-resolution broadband spectroscopic module could be realized with a heterodyne system incorporating QC lasers and detectors.

## Figures and Tables

**Figure 1 sensors-21-05706-f001:**
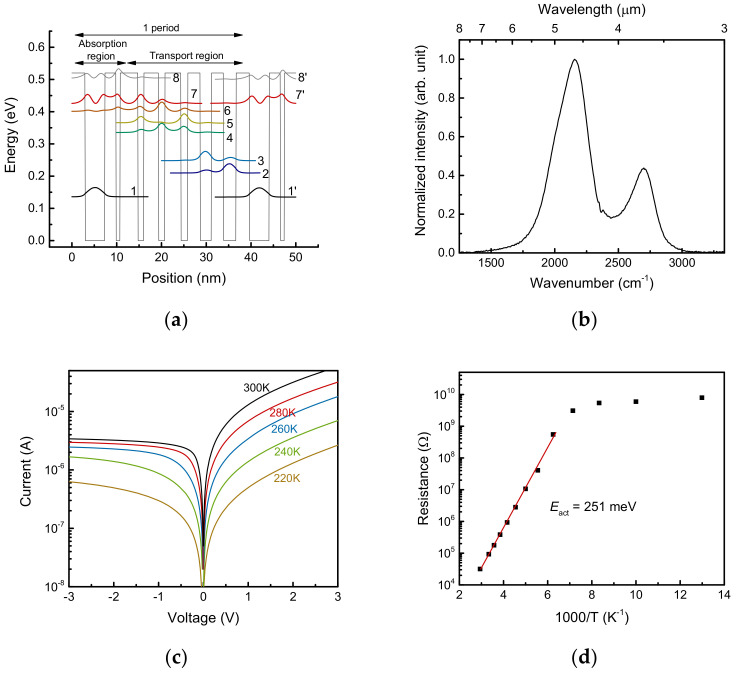
(**a**) Schematic of the conduction band and moduli squared of the relevant wavefunctions of the quantum cascade detector. The In_0.53_Ga_0.47_As/In_0.52_Al_0.48_As layer sequence of one period of the active regions, in angstroms, starting from the absorption well, is 44/**25**/9/**40**/13/**33**/14/**36**/15/**28**/25/**27**/28/**30**, where InAlAs barrier layers are in bold, InGaAs quantum-well layers are in roman type, and the doped layer (Si, 4 × 10^17^ cm^−3^) is underlined. (**b**) Photoresponse spectrum of the device measured without cooling. (**c**) Dark current–voltage characteristics over the temperature range 220–300 K. (**d**) Arrhenius plot of the measured differential resistance. The red line is the fit of the Arrhenius model, *L* = Aexp[*E*_act_/*k*_B_*T*], where *L* is lifetime, A is a constant, *E*_act_ is the activation energy, *k*_B_ is Boltzmann’s constant, and *T* is the device temperature.

**Figure 2 sensors-21-05706-f002:**
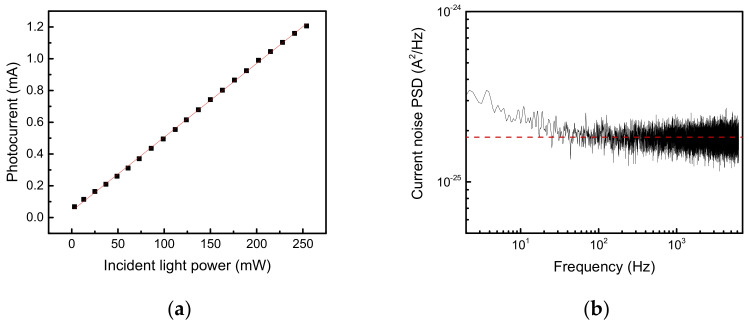
(**a**) Measured photocurrent vs. incident light power of the 2220-cm^−1^ distributed-feedback quantum cascade (QC) laser. The measured results are plotted as square black dots, and the red line is a linear fit. (**b**) Current noise power spectrum density of the QC detector at room temperature (no cooling). The red dashed line indicates the Johnson–Nyquist noise level calculated from 4*k*_B_*T*/*R*, where *k*_B_ is Boltzmann’s constant, *T* is the device temperature, and *R* is the device resistance.

**Figure 3 sensors-21-05706-f003:**
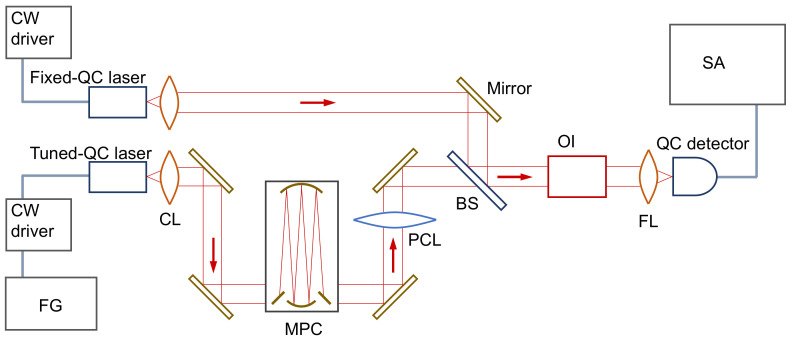
Schematic of heterodyne spectroscopy. FG: function generator, MPC: multi-pass cell, CL: collimating lens (ZnSe, aspheric, working distance of 3 mm), PCL: plano-convex lens (CaF_2_, focal length of 300 mm), FL: focusing lens (ZnSe, aspheric, working distance of 1 mm), BS: beam splitter, OI: optical isolator, and SA: spectrum analyzer.

**Figure 4 sensors-21-05706-f004:**
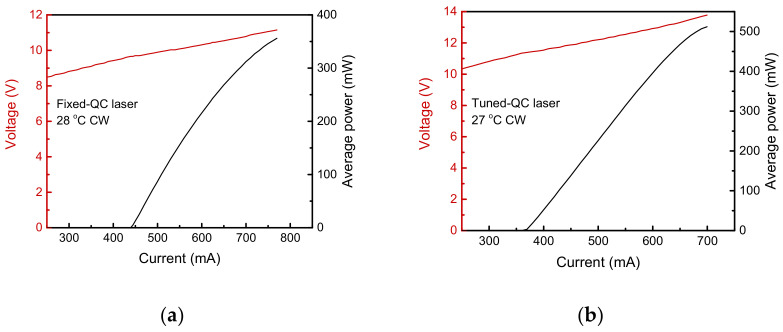
Continuous-wave current–voltage–light output characteristics of (**a**) the fixed quantum cascade (QC) laser operated at the heatsink temperature of 28 °C and (**b**) the tuned QC laser operated at the heatsink temperature of 27 °C.

**Figure 5 sensors-21-05706-f005:**
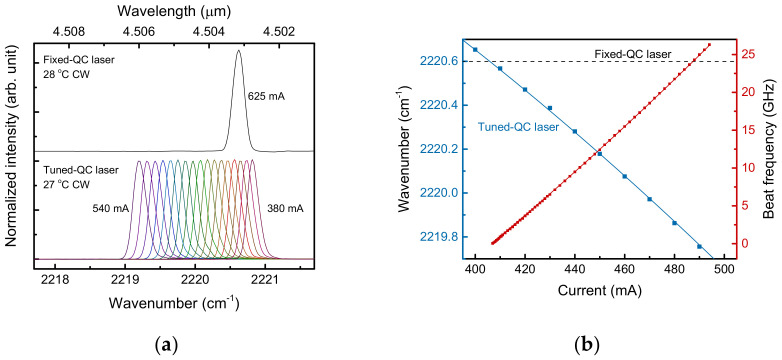
Emission wavelengths of the quantum cascade (QC) lasers and the frequency of the generated beat signal. (**a**) The spectra of the QC lasers depend on the injection current. The upper spectrum corresponds to the fixed QC laser with a fixed current of 625 mA, and the lower spectra correspond to the current-tuned QC laser. (**b**) The beat frequency (red square dots) measured with a QC detector and spectrum analyzer. The red line is a prediction of the beat frequency calculated from the wavelength difference of the fixed (dashed line) and tuned QC lasers (blue square dots). The blue line is the fit.

**Figure 6 sensors-21-05706-f006:**
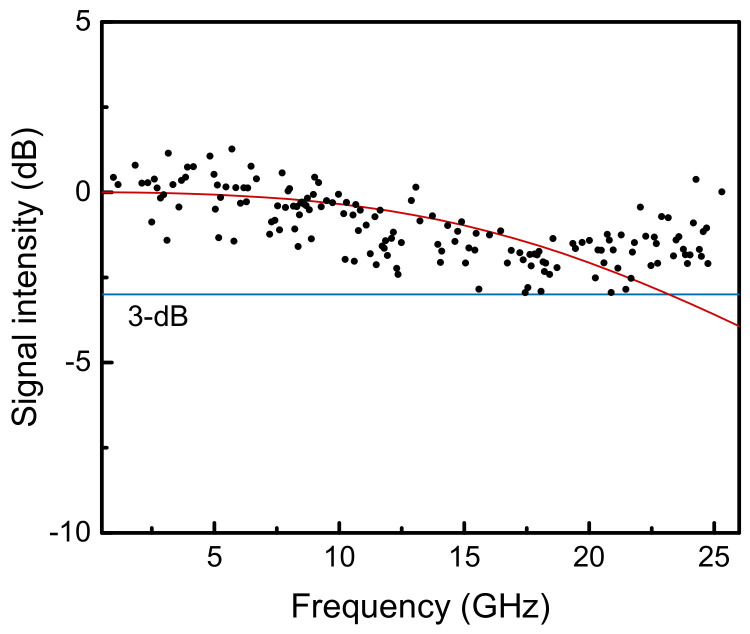
Frequency response of the quantum cascade (QC) detector observed in heterodyne beat measurement. The black dots are experimental data obtained by using the max-hold trace mode of the spectrum analyzer. The red curve is a theoretical prediction of the frequency response using the measured capacitance of 0.19 pF, the calculated inductance of 0.21 nH, and the input impedance of 50 Ω.

**Figure 7 sensors-21-05706-f007:**
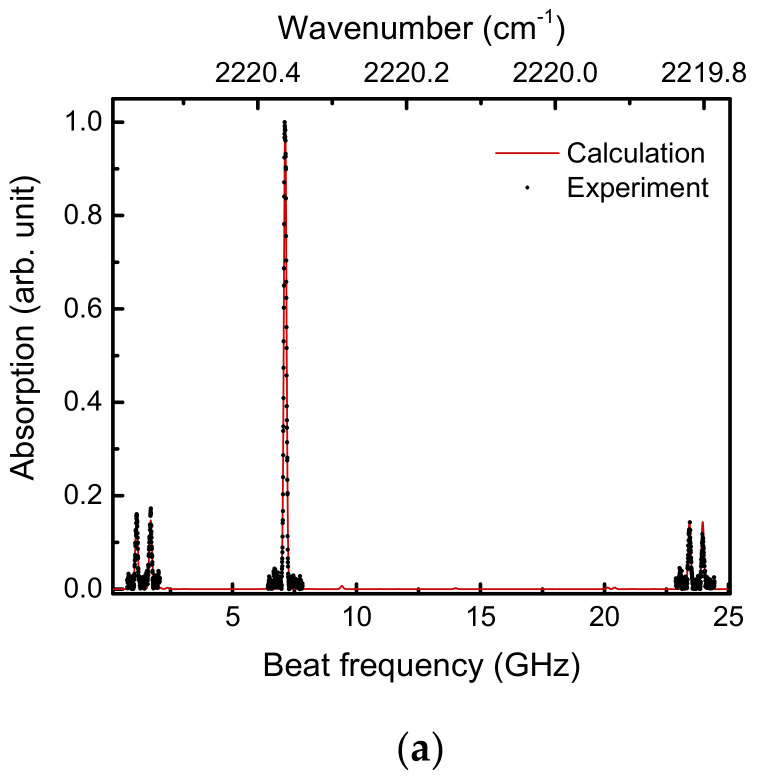
N_2_O absorption lines observed with broadband heterodyne spectroscopy. The experimental data are dots, and the red lines are calculated absorptions of N_2_O based on SpectraPlot. The experimental and calculated spectra were normalized by maximum values in the determining areas. (**a**) Entire frequency range. (**b**–**d**) Enlarged absorption spectra corresponding to the beat frequency domains of 0.78–1.98 GHz, 6.47–7.67 GHz, and 23.1–24.3 GHz, respectively.

## Data Availability

The data presented in this study are available from the corresponding author upon reasonable request.

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
