# Peer review of "Application of High-Speed Quantum Cascade Detectors for Mid-Infrared, Broadband, High-Resolution Spectroscopy"

_sensors, 2021, doi:10.3390/s21175706_

Round 1

Reviewer 1 Report

A high-speed quantum cascade (QC) detector is developed in the manuscript for broadband, high-resolution, heterodyne, mid-infrared absorption spectroscopy. The performance of the detector was characterized and verified by N2O absorption experiment. Some suggested revision are as follows.

1. In the introduction, the authors should clarify the state of the art of MIR QC detectors, the advantages and disadvantages compared with traditional MIR detectors, and the research gap, focusing on the characteristics of QC detector designed in this paper.

Relevant literature can be cited, e.g.

[1]Piotrowski, A.; Piotrowski, J. Uncooled Infrared Detectors in Poland, History and Recent Progress. In Proceedings of the 26th European Conference on Solid-State Transducers (Eurosensors), Krakow, Poland,9–12 September 2012; pp. 1506–1512

[2]Du Z, Zhang S , Li J , et al. Mid-Infrared Tunable Laser-Based Broadband Fingerprint Absorption Spectroscopy for Trace Gas Sensing: A Review[J]. Applied Sciences, 2019, 9(2):338.

2. How the resolution bandwidth (5MHz) is determined in the manuscript?

3. Is the 2GHz scanning range a static parameter? At 1 kHz scanning frequency, the tuning range may become smaller.

4. How is the N2O gas pressure (50Pa) to be guaranteed in the experiment?

5. In section 4, When EC-QCL or OFC is used for broadband measurement, is it necessary to correct the unevenness of the light intensity response of the QC detector, i.e. Fig. 1 (b)?

Reviewer 2 Report

----------------------------------------------------------------------
Report of the Referee -- 
----------------------------------------------------------------------

This work is well within the scope of Sensors. It shows 
an introductory background material, sufficient for someone not an expert 
in this area to understand the context and significance of this work, 
with good references to follow. The authors have shown that it is possible
broadband, high-resolution, heterodyne, mid-infrared absorption spectroscopy was performed with a high-speed quantum cascade (QC) detector. In particular, high-speed operation with a 3-dB bandwidth over 20 GHz was realized in an un-262 cooled QC detector with a 4.6-um peak response.
I find several shortcomings that need to be addressed before
the manuscript is suitable for publication:
(1) Are there any previous published data proving the validity of 
high-speed quantum cascade (QC) detector? Please discuss. 
(2) The first paragraph in the second chapter was superficially written.
I have opted to recommend a Minor Revision for the current version of the article.
